# Formation of Multiscale Pattern Structures by Combined Patterning of Nanotransfer Printing and Laser Micromachining

**DOI:** 10.3390/nano13162327

**Published:** 2023-08-13

**Authors:** Tae Wan Park, Young Lim Kang, Eun Bin Kang, Seungmin Kim, Yu Na Kim, Woon Ik Park

**Affiliations:** 1Department of Materials Science and Engineering, Pukyong National University (PKNU), Busan 48513, Republic of Korea; twpark0125@gmail.com (T.W.P.); ylkang0914@gmail.com (Y.L.K.); beiunn803@gmail.com (E.B.K.);; 2Department of Materials Science and Engineering, Korea University, Seoul 48513, Republic of Korea; 3RanoM R&D Center, RanoM Co., Ltd., Busan 48548, Republic of Korea; sm.kim131@gmail.com

**Keywords:** multiscale, nanostructure, nanotransfer printing, laser micromachining

## Abstract

Various lithography techniques have been widely used for the fabrication of next-generation device applications. Micro/nanoscale pattern structures formed by lithographic methods significantly improve the performance capabilities of the devices. Here, we introduce a novel method that combines the patterning of nanotransfer printing (nTP) and laser micromachining to fabricate multiscale pattern structures on a wide range of scales. Prior to the formation of various nano-in-micro-in-millimeter (NMM) patterns, the nTP process is employed to obtain periodic nanoscale patterns on the target substrates. Then, an optimum laser-based patterning that effectively engraves various nanopatterned surfaces, in this case, spin-cast soft polymer film, rigid polymer film, a stainless still plate, and a Si substrate, is established. We demonstrate the formation of well-defined square and dot-shaped multiscale NMM-patterned structures by the combined patterning method of nTP and laser processes. Furthermore, we present the generation of unusual text-shaped NMM pattern structures on colorless polyimide (CPI) film, showing optically excellent rainbow luminescence based on the configuration of multiscale patterns from nanoscale to milliscale. We expect that this combined patterning strategy will be extendable to other nano-to-micro fabrication processes for application to various nano/microdevices with complex multiscale pattern geometries.

## 1. Introduction

High-resolution patterning technologies have become essential for the development of various next-generation device systems that require periodic micro and nanoscale pattern structures, such as high-sensitivity sensors [1,2], nonvolatile memory devices [3,4,5], batteries [6,7], energy harvesters [8,9,10], and electric vehicles [11,12,13]. Micro-to-nanoscale regimes, including atomic clusters, nanowires, nanoribbons, nanoparticles, and layered lamellar films, allow the significant enhancement of output performance in device systems with complex circuits [14,15]. Micro- and nanostructured materials can be representatively generated through advanced fabrication processes, such as chemical synthesis, the bottom-up lithographic method, and physical process-based top-down manufacturing, which becomes more significant when integrated into device systems [14,16]. For these reasons, various research groups have consistently investigated and developed novel micro/nanofabrication techniques based on physical and chemical processes [17,18,19].

For example, a nanosecond laser lithography method is utilized to fabricate battery cells with ultra-high areal capacities in the form of surface-patterned periodic concave-convex microstructures on zinc metal battery anodes [20]. This approach imparts hydrophilic and zincophilic surfaces of zinc foil and achieves an unprecedented high areal capacity of 7.6 mA h cm^−1^. The electrode of flexible nano- and micro-optoelectronic devices can be successfully patterned to arbitrary geometries by directly integrating multiple organic crystals by photolithography [21]. The compatibility of the reported photolithography method with organic molecular crystals will pave the way toward a key idea for the development of large-scale optoelectronic device applications. As a typical large-area patterning method, roll-to-roll (R2R) nanoimprint lithography (NIL) can effectively create nano/microscale pattern structures due to its excellent throughput and process simplicity [22]. The advanced NIL methods have been consistently developed and reported in terms of large-area patterning with industrial device manufacturing [23,24].

Above these useful techniques, several emerging bottom-up and top-down lithographic methods, such as pattern transfer printing [25,26,27,28,29,30,31], femtosecond laser lithography [32,33,34], electron-beam lithography (EBL) [35,36], dip-pen nanolithography (DPN) [37,38], extreme ultraviolet (EUV) lithography [39,40], advanced imprint lithography [41,42,43,44,45], and the directed self-assembly (DSA) of block copolymers (BCPs) [46,47,48,49,50,51] have also been widely used and developed for the realization of precisely controlled pattern shapes comparable to those created with other lithography techniques. However, in general, most lithography processes developed thus far have struggled to create multiscale pattern features, such as nano-in-micro, nano-to-micrometer, and/or nano-to-micro-to-millimeter structures on the various target substrates using a single lithographic process, because these patterning techniques mostly require complex, plasma-based dry etching processes in a vacuum chamber to obtain periodic, complex structures. Another reason is that the target resolution range, which is mainly applied to each process, is determined beforehand in the manufacturing systems. In order to diversify and hybridize the pattern regimes, a combination of patterning methods with different merits is required.

Here, we introduce a useful and practical multiscale pattern formation method realized by a patterning method that combines thermally assisted nanotransfer printing (T-nTP) and laser micromachining. We demonstrate the pattern formation of nanoscale Au line structures on colorless polyimide (CPI) film by the T-nTP process as an initial substrate on which to realize various multiscale patterns. Then, the substrate-tailored patterning conditions of micro-in-millimeter (MIM) structures with high patternability on a variety of soft and rigid surfaces are experimentally established. Furthermore, we demonstrate how to obtain well-organized nano-in-micro-in-millimeter (NMM) structures on a CPI substrate, and present complex holographic text-shaped NMM patterns that show a noticeable effect of the desired structures based on improved visibility depending on the luminescence of the nanopattern-embedded microscale units.

## 2. Materials and Methods

### 2.1. Formation of Periodic Nanoscale Patterns

To form well-ordered nanoscale pattern structures on a flexible and transparent CPI substrate, the T-nTP technique, capable of generating periodic micro-to-nanoscale pattern structures, was employed. A Si master mold with both a width and a space of 250 nm was fabricated by conventional KrF photolithography and reactive ion etching (RIE) processes. A spin-casted positive photoresist (PR, Dongjin Semichem Co., Ltd., Seoul, Republic of Korea) with 400 nm on the Si wafer was exposed using a KrF scanner (Nikon, NSR-S203B) followed by a developing step using a developer solution (tetramethylammonium hydroxide, Dongjin Semichem Co., Ltd.). The remaining separated PR patterns were used as a mask in the etching step to fabricate the patterned surface of the Si wafer with plasma treatment (gas: CF_4_, working pressure: 7 mTorr, etching power: 250 W). After the removal of the residual PR pattern, we finally obtained nanoscale pattern structures on the surface of the Si wafer. Prior to the replication step, a surface of a Si mold was modified by a brush treatment using hydroxyl-terminated polydimethylsiloxane (PDMS-OH, Polymer Source Inc., Dorval, QC, Canada) with a molecular weight (MW) of 5 kg/mol at 150 °C for 2 h in a vacuum oven. PDMS brush-treated surface allows easy separation of materials to be formed on the Si mold. A poly(methyl methacrylate) (PMMA, MW = 120 kg/mol, Sigma Aldrich Inc., St. Louis, MO, USA) solution at 3.5 wt% in a mixture of toluene, acetone, and heptane (4:4:2 by volume) was spin-casted onto the Si mold at 5000 rpm for 23 s. The coated PMMA thin film was manually attached and detached with a mild pressure (less than 15 kPa) using an adhesive polyimide (PI) tape. A functional material, in this case, Au, was deposited on the replicated PMMA pattern by a magnetron ion sputter coater (MSP-1S, Vacuum Device Inc., Japan) with a thickness of 20 nm. The T-nTP process was then performed using a heat-rolling-press system (LAMIART-470 LSI, GMP Corp.) capable of providing both uniform pressure and heat after ensuring contact between the Au/PMMA layer and the CPI substrate. The printing temperature and speed for optimum pattern transfer printing in this study were set to 150 °C and 500 mm/min. The adhesive PI tape used during this step was manually removed from the CPI substrate, after which the residual PMMA pattern was removed by washing with a solvent (toluene) and with a reactive ion etching (gas: O_2_, flow rate of the gas: 30 standard cubic centimeters per minute, working pressure: 15 mTorr, plasma power: 60 W, and etching time: 20 s).

### 2.2. Laser Micromachining

Precise laser patterning with a fiber laser marker (Cat-Fs Series, Marcs Co., Ltd., Tokyo, Japan), which can produce a resolution of ~40 µm, was implemented to obtain the designed micro-to-millimeter scale pattern structures on surfaces of various substrates. The laser beam of the ytterbium fiber laser marker can precisely produce surface-patterned products at a maximum speed of 10,000 mm/s with 100 W power. The MIM and NMM patterns formed on the surface of various substrates were systematically manufactured under the substrate material-tailored optimum power and speed conditions (soft PMMA thin film: 7 mW and 1000 mm/s, rigid poly(ethylene terephthalate) (PET) and CPI films: 10 mW and 1000 mm/s, stainless steel (SUS) or metal plate: 30 W and 500 mm/s, and Si substrate: 75 W and 800 mm/s).

### 2.3. Characterization

The multiscale pattern structures, including printed Au nano lines and laser patterned micro- and millimeter-scale units, were observed using a field emission scanning electron microscope (FE-SEM, MIRA3, TESCAN) under the operating conditions with an acceleration voltage of 10 kV and a working distance of 8 mm.

## 3. Results

### 3.1. Combined Patterning with Nanotransfer Printing and Laser Micromachining

The conceptual strategy used to fabricate NMM pattern structures is schematically depicted in Figure 1. Figure 1a shows the process sequence of NMM pattern formation by the combined patterning method of T-nTP and laser micromachining. Periodic nanoscale patterns are formed on the target substrate by the T-nTP process. Laser micromachining is then implemented on the initial substrates of transfer-printed nanopatterns. During the laser patterning step, the laser beam moves quickly at a speed of 500~10,000 mm/sec and engraves the surface of the target substrate based on pre-designed two-dimensional (2D) images. Figure 1b presents an example of a detailed NMM structure that includes multiscale patterns of nanoscale, microscale, and millimeter-scale pattern structures. Several microscale arrays are located inside the millimeter-scale one-unit cell, and numerous nanoscale patterns are embedded in one microscale pattern unit.

### 3.2. Periodic Nanoscale Pattern Formation by T-nTP

To create nanoscale patterns on an arbitrary substrate, a pattern transfer printing process capable of effectively generating periodic micro-to-nano geometries was used. Figure 2 shows the formation of nanoscale pattern structures on a flexible and transparent CPI film realized by the T-nTP method. The procedure for fabricating nanoscale patterns with functionality is schematically shown in Figure 2a. First, a Si master mold is fabricated by conventional photolithography. A PMMA layer is then coated onto the Si mold and is subsequently peeled off using an adhesive PI tape. PMMA patterns with the reversed shape of the surface of the Si mold are thus successfully replicated. An Au material is then deposited on the protruding parts of the PMMA replica pattern by angle deposition followed by contact on the target substrate. After the transfer printing process, an adhesive PI tape and replica layer are manually removed by hand and solvent flow washing, respectively. Figure 2b shows a cross-section view SEM image of the Si master mold with a width/space of 250 nm. High-quality patterns were precisely engraved on the Si substrate through the photolithography process. Figure 2c presents photographic and top-view SEM images of transfer-printed Au patterns with a width/space of 250 nm, showing precise, well-ordered nanoscale line structures created by the soft printing technique of T-nTP. During the contact-based T-nTP process, regular perfect pressure to avoid air bubble defects and uniform heat injection to effectively move functional nanostructured materials are very important. A heat-rolling-press system can provide uniform pressure and heat on the patterning area for successful pattern transfer printing. PMMA replica material typically undergoes thermal expansion when applying heat above a certain range. However, in the case of a spin-coated porous PMMA layer from the liquid PMMA solution made by dissolving solid PMMA in the liquid solvent, it may shrink, passing the heated rolls depending on the porosity of the polymer layer, as we previously reported [26]. The interface between the PI tape and the PMMA replica layer can be easily separated due to the weakening of the sticky layer of the PI tape by heat injection. According to this principle of T-nTP, we successfully printed Au line patterns with a width of 250 nm on a flexible CPI substrate over an area of 1.2 cm × 1.2 cm. The inset shows a fast Fourier transform (FFT) pattern obtained from the SEM image, clearly indicating well-ordered high-resolution line structures.

### 3.3. MIM Pattern Formation by Laser Micromachining

To confirm the reliability of the laser patterning process when used on the target substrates, we implemented substrate-tailored laser micromachining on various surfaces. Figure 3 shows the formation of MIM pattern structures by laser micromachining. The sequential laser process of millimeter and micrometer machining is schematically illustrated in Figure 3a. A laser beam can unrestrictedly move to any angle and/or in any direction, including 0° (*x*-axis), 45°, and 90° (*y*-axis), on the substrate. Figure 3b displays a photographic image of the progress of laser patterning on a non-patterned pristine CPI substrate. Based on the optimum laser power condition (see Methods for more details), we systematically designed and successfully manufactured MIM pattern structures with an area of 1 cm × 1 cm, as shown in Figure 3c. The areas shown in blue and red represent the designated micromachining zones for the formation of millimeter-scale and micrometer-scale patterns, respectively. After the second micromachining step, multiple µm-scale unit cells were regularly arranged inside the one mm-scale unit cell. The sides of square-shaped µm-scale units (81 cells) and mm-scale units (9 cells) are 350 µm and 2 mm, respectively. The laser beam-based processing presented in this study enabled reliable micro-scale patterning. It should be noted that the um-scale pattern structures for the realization of plasmonic metasurfaces or optical device applications can also be manufactured by other programmable beam shaping, such as electron beam lithography [52,53] and femtosecond laser lithographic process [54,55]. Although the ytterbium fiber laser-based micromachining system used in this study can achieve a resolution of ~40 µm, we designed unit cells with 350 µm size because the purpose of the fabrication in this study was to make a micro-patterned area that can be visible to the naked eye. Figure 3d shows MIM patterns formed on surfaces of various substrates. We also assessed laser patterning on soft PMMA thin film, rigid PET film, SUS plate, and Si substrate, resulting in MIM structures with excellent resolutions. These results imply that laser micromachining realized by the laser beam of a ytterbium fiber laser marker can create micro-to-milliscale pattern structures on various surfaces by means of a precisely controlled process with optimum laser power and speed conditions.

### 3.4. Fabrication of Various NMM Structures on Flexible Substrate

Figure 4 shows the formation of various NMM structures realized by the combined patterning method of T-nTP and laser micromachining. The conceptual strategy of the NMM structure is that periodic nanoscale pattern structures are included in a micrometer-scale unit, and these multiple microscale unit cells with nanoscopic patterns are embedded inside a millimeter-scale unit cell, as schematically shown in Figure 4a. Figure 4b shows square-shaped NMM pattern structures. We systematically engineered and performed the patterning of microscale square and milliscale square units with sides measuring 350 µm and 2 mm in size, respectively. Figure 4c shows dot-shaped NMM patterns, which include microscale dots and milliscale dot units with diameters of 400 µm and 2.5 mm, respectively. Nanoscale Au line patterns formed by T-nTP were inherent in all microscale square and dot structures, as shown in the magnified SEM images in Figure 4b,c. These figures clearly indicate that the combined patterning method of T-nTP and laser micromachining can effectively generate various multiscale NMM structures. Here, it should be noted that this approach allows various imaginable NMM structures to be formed on flexible and transparent substrates under room temperature (RT) conditions. We also created text-shaped NMM structures, as shown in Figure 4d. The effect of luminescence on the microscale units with nanopatterns significantly enhances the visibility of millimeter-scale texts, imparting a more noticeable effect on the desired structures. On the basis of these results, the NMM pattern structures realized by the combined micro/nanopatterning methods can be applied to various areas, such as advanced metasurface holography, surface engineering, and product logos with good aesthetic sensibility. In addition, this combined patterning strategy for the realization of high-resolution multiscale pattern geometries that cannot be achieved with a single process will be applicable to various device systems that require complex nano-to-microstructures within device systems.

## 4. Conclusions

In summary, a facile and useful patterning method that combines T-nTP and laser micromachining to obtain multiscale NMM pattern structures was proposed. We fabricated highly ordered nanoscale Au line patterns on CPI film by the T-nTP process as an initial substrate for the formation of targeted NMM structures. We demonstrated the excellent patternability of the laser lithography process on various surfaces, showing substrate-tailored patterning results of MIM structures on diverse substrates, specifically spin-coated soft PMMA film, rigid PET film, SUS plate, and Si substrate. Well-defined NMM structures clearly showing periodic Au line patterns with a width of 250 nm in the form area of micro/millimeter-scale square patterns were successfully obtained by employing the sequential lithography methods of T-nTP and laser patterning. We also showed the pattern generation of dot-shaped NMM structures consisting of transfer-printed Au lines with a width of 250 nm, microdots with a diameter of 400 µm, and millimeter-scale dots with a diameter of 2.5 mm by the combined patterning strategy. Furthermore, we demonstrated specially designed text-shaped NMM pattern structures that exhibited a holographic effect on the structures used based on enhanced visibility stemming from the luminescence of the microscale units. These results not only demonstrate the excellent patternability of the combined lithography method to obtain complex multiscale structures but also provide new pathways toward surface engineering related to metasurface-based holographic systems and/or advanced photonics.

## Figures and Tables

**Figure 1 nanomaterials-13-02327-f001:**
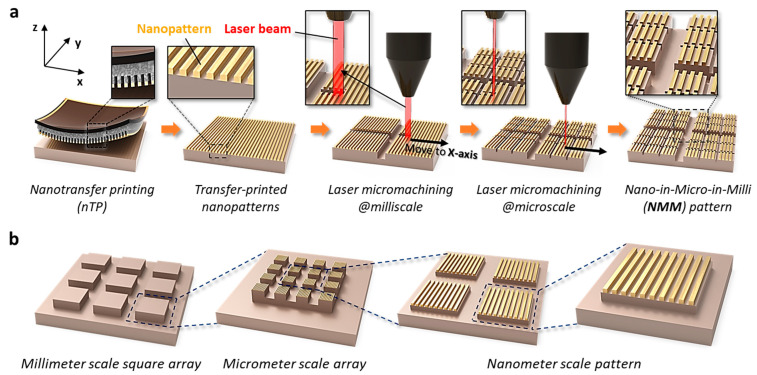
Conceptualization for the fabrication of NMM pattern structures. (**a**) The process sequence for the formation of the NMM pattern structure by the combined patterning of thermally assisted nanotransfer printing (T-nTP) and laser micromachining. Periodic nanoscale line structures are transferred on the target substrate via the T-nTP process. A laser patterning process is then implemented on transfer-printed nanostructures. (**b**) Schematic illustrations of a patterned NMM structure by the combined patterning strategy of T-nTP and laser patterning methods. The NMM-patterned structure includes highly ordered nano-to-micro-to-millimeter scale multi-pattern structures.

**Figure 2 nanomaterials-13-02327-f002:**
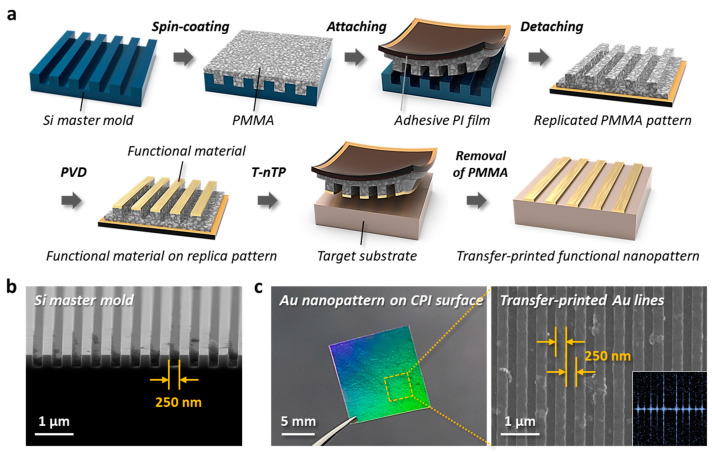
Formation of periodic nanoscale pattern structures via T-nTP. (**a**) The process sequence of nanoscale pattern formation by the T-nTP method. Functional line/space nanopatterns are transferred using a replicated PMMA film from a Si master mold fabricated by conventional photolithography. (**b**) SEM image of the Si master mold with a line width/space of 250 nm. (**c**) Photographic and SEM images of transfer-printed Au nanopatterns with a width/space of 250 nm on CPI substrate, showing well-defined nanoscale line structures.

**Figure 3 nanomaterials-13-02327-f003:**
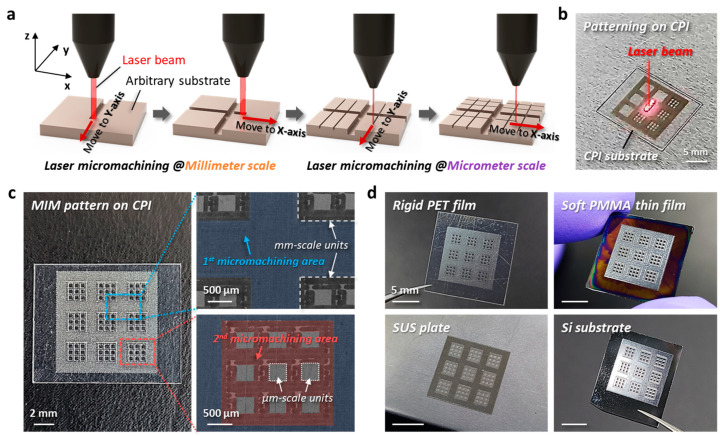
Formation of MIM pattern structures by laser micromachining. (**a**) Schematic illustrations of two steps of the laser micromachining with millimeter and micrometer processes. (**b**) Photographic image of the formation of the MIM pattern on a CPI substrate using a laser beam during the micropatterning process. (**c**) The MIM structure patterned by the multiple laser machining system. Nine units of a microscale square array with a width of 350 µm are obtained in the 2 × 2 mm^2^ square array. (**d**) MIM structures on various surfaces, specifically rigid PET film, soft PMMA film, SUS plate, and Si substrate.

**Figure 4 nanomaterials-13-02327-f004:**
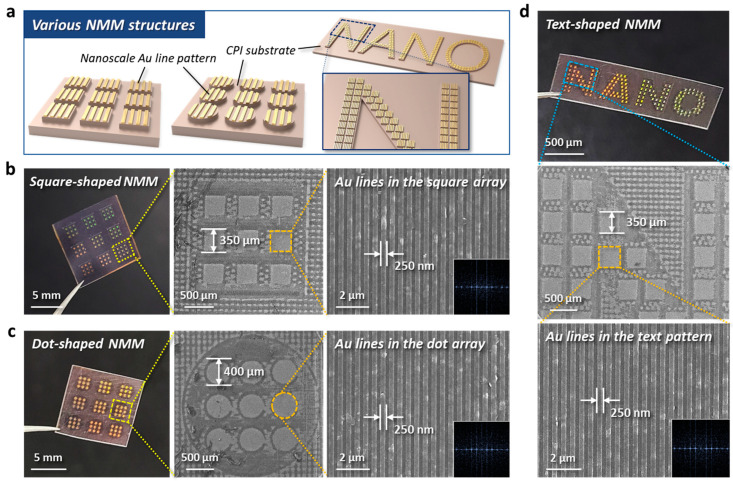
Various multiscale NMM pattern structures on flexible CPI substrates. (**a**) Schematically depicted NMM structures created by the combined patterning method of T-nTP and laser micromachining. (**b**) Square-shaped NMM pattern structures consisting of µm-scale squares with a side of 350 µm and mm-scale squares with a side of 2 mm. (**c**) Dot-shaped NMM pattern structures consisting of µm-scale dots with a diameter of 400 µm and mm-scale dots with a diameter of 2.5 mm. (**d**) Text-shaped NMM structures. All NMM patterns contain Au lines with a width of 250 nm within the microscale units, in this case, squares, dots, and texts. The multiscale NMM structures reveal better visibility on the desired millimeter-scale structures stemming from the luminescence of the micro units with Au nanopatterns.

## Data Availability

The data presented in this study are available on request from the corresponding author.

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
