# Peer review of "Formation of Multiscale Pattern Structures by Combined Patterning of Nanotransfer Printing and Laser Micromachining"

_nanomaterials, 2023, doi:10.3390/nano13162327_

Round 1

Reviewer 1 Report

The manuscript entitled "Formation of Multiscale Pattern Structures by Combined Patterning of Nanotransfer Printing and Laser Micromachining” by Park et al. describes a technique that can be applied for multiscale patterning of a substrate. The technique is basically a combination of two methods: Thermally assisted nanotransfer printing followed by laser machining. The authors have demonstrated multiscale printing on polymer, stainless steel and Silicon. The developed technique is unique and versatile. Hence, I recommend the manuscript for publishing in Nanomaterials Journal with a minor revision. I do not see Fig. 3 in the manuscript. The authors are therefore requested to insert the corresponding figure in the manuscript.

Author Response

[General review] The manuscript entitled "Formation of Multiscale Pattern Structures by Combined Patterning of Nanotransfer Printing and Laser Micromachining” by Park et al. describes a technique that can be applied for multiscale patterning of a substrate. The technique is basically a combination of two methods: Thermally assisted nanotransfer printing followed by laser machining. The authors have demonstrated multiscale printing on polymer, stainless steel and Silicon. The developed technique is unique and versatile. Hence, I recommend the manuscript for publishing in Nanomaterials Journal with a minor revision. I do not see Fig. 3 in the manuscript. The authors are therefore requested to insert the corresponding figure in the manuscript.

 [Our response] We greatly appreciate the reviewer's positive and kind evaluation of our research paper. Figure 3 was included in the manuscript at the submission stage, but it seems to have been deleted, as the reviewer mentioned. In response to the reviewer, we inserted the corresponding Figure 3 in the revised manuscript.

 [Modification of the manuscript]

Figure 3. Formation of MIM pattern structures by laser micromachining. (a) Schematic illustrations of two-steps of the laser micromachining with millimeter and micrometer processes. (b) Photographic image of the formation of MIM pattern on a CPI substrate using a laser during the micropatterning process. (c) MIM structure patterned by the multiple-laser machining system. Nine units of a microscale square array with a width of 350 µm are obtained in the 2 × 2 mm2 square array. (d) MIM structures on various surfaces, specifically rigid PET film, soft PMMA film, SUS plate, and Si substrate. on line 206 of the revised manuscript.

Reviewer 2 Report

Generally, this is very well written paper, the scientific content is extremely clear and easy to follow, and I have just a few comments. The application of using multiple approaches for achieving orders of magnitude variations in feature sizes is interesting. However, my expertise is in laser micromachining, and hence I am not able to comment on the novelty or quality of the processes itself. I hope that you are able to use other reviewer comments to help on this.

In your introduction, your main novelty seems to be stated as “In general, most lithography processes developed thus far have struggled to create multiscale pattern features using a single process.” But in your work, in the figures, you do use multiple steps. I’m not completely sure what your novelty here is therefore, and perhaps you can more precisely define this in the introduction section?

Figure 2 – the FFT spectrum is very small, and doesn’t show the frequencies of the peaks. Can you make this FFT spectrum clearer and more useful to the reader?

The fibre laser is 100W. Given you are trying to achieve high resolution machining, what was the motivation for using a fibre laser and not a femtosecond laser system? Likewise, rather than using a scanning beam, there are other options for spatially shaped beams. There is a vast literature on SLMs and DMDs for beam shaping, particularly with femtosecond laser machining. It might therefore seem quite important to compare the resolutions that you achieve, with the existing literature in this area?

Can you talk about some of the manufacturing applications here, and state how your fabrication method unlocks new capability?

Specifically, what samples/features does your fabrication approach allow, that currently is not achievable by other methods? I don’t think I saw a comparison of resolutions and size scales, and fabrication speed, etc. that compares your work with existing work in the literature.

Author Response

[General review] Generally, this is very well written paper, the scientific content is extremely clear and easy to follow, and I have just a few comments. The application of using multiple approaches for achieving orders of magnitude variations in feature sizes is interesting. However, my expertise is in laser micromachining, and hence I am not able to comment on the novelty or quality of the processes itself. I hope that you are able to use other reviewer comments to help on this.

[Our response] We appreciate the reviewer’s positive and thoughtful evaluation of our research paper. This article focuses on the formation of multiscale pattern structures by a novel combined patterning method of nanotransfer printing and laser micromachining. We are developing several combined lithographic strategies for effective fabrication of nano-to-microscale pattern structures, which can be applied to various next-generation device systems. We are also studying and making an effort to realize these strategies with a simple and practical process as the reviewer mentioned. In addition, we agree with the reviewer’s concern regarding the quality of the laser micromachining process itself. Indeed, this study suggests that the combined nanopatterning of nTP and laser patterning is the first attempt to effectively fabricate nano-to-micro-to-millimeter pattern structures. We also trying to develop new and useful combined nanopatterning methods, as described below. The use of functional material which shows excellent quality multiscale patterns also can be patterned by laser-based patterning process in near future studies.

Figure. Combined patterning method of block copolymer self-assembly and nanotransfer printing. (a-c) Direct printing of ultrathin block copolymer film with nano-in-micro patter structures. (d) Transfer-printed nano-in-micro patterns. (Unpublished work)

Figure. Pattern formation outcomes via the combined patterning method of block copolymer self-assembly and nanotransfer printing. (a) Surface plotted 3D images of complex nut-shaped micropattern consisting of sub-20 nm SiOx nanodots. (b) Procedure for the pattern formation of nano-in-micro multiscale pattern structures at the eight-inch wafer scale. (Unpublished work)

Figure. High-resolution nanotransfer printing of porous crossbar array using patterned metal molds by imprint lithography. (a) Process sequence of nanoscale pattern formation using imprinted metal molds. (b) Surface-patterned metallic molds by extreme-pressure imprint lithography. (c) Replication of PMMA line patterns obtained from imprinted metal molds. (d) Various functional line structures and nanoporous crossbar array by the combined patterning method of hard imprinting and transfer printing processes. (Unpublished work)

[Comment 1] In your introduction, your main novelty seems to be stated as “In general, most lithography processes developed thus far have struggled to create multiscale pattern features using a single process.” But in your work, in the figures, you do use multiple steps. I’m not completely sure what your novelty here is therefore, and perhaps you can more precisely define this in the introduction section?

[Our response 1] We appreciate the reviewer’s helpful comments. We agree with the reviewer that we did not explain the process simplicity of the suggested combined nanopatterning to generate multiscale patterns compared to the conventional lithographic methods. Conventional lithographic methods such as photolithography, nanoimprint, and directed self-assembly of block copolymers can generate high-resolution patterns. However, they have struggled to achieve pattern formation of complex multiscale structures (nano-to-micrometer structures and/or nano-to-micro-to-millimeter structures on the arbitrary substrates) when using a single lithographic process. Because these typical patterning techniques necessarily require complex, plasma-based dry etching processes in vacuum chamber to obtain periodic, complex structures. In particular, the second or additional patterning process is also needed to obtain complex nano-in-micro-patterns, which cannot be obtained by conventional patterning-based processes. So, we suggested a combined patterning method of nTP and laser micromachining to easily produce multiscale pattern geometries. In response to the reviewer’s helpful comments, we added the following sentence to explain the novelty of this research, as the reviewer suggested.

[Modification of the manuscript]

“However, in general, most lithography processes developed thus far have struggled to create multiscale pattern features, such as nano-in-micro, nano-to-micrometer and/or nano-to-micro-to-millemeter structures on the various target substrates using a single lithographic process. Because these patterning techniques mostly require complex, plasma-based dry etching processes in vacuum chamber to obtain periodic, complex structures. Another reason is that the target resolution range, which is mainly applied to each process, is determined beforehand in the manufacturing systems.” on line 63 of the revised manuscript.

[Comment 2] Figure 2 – the FFT spectrum is very small, and doesn’t show the frequencies of the peaks. Can you make this FFT spectrum clearer and more useful to the reader?

[Our response 2] We appreciate the reviewer’s helpful comment and agree with the concern regarding the unclear FFT spectrum data. As the reviewer suggested, we revised FFT patterns in Figure 2 to be clearer and bigger for readers to easily understand the pattering results.

[Modification of the manuscript]

Figure 2. Formation of periodic nanoscale pattern structures via T-nTP. on line 206 of the revised manuscript.

[Comment 3] The fibre laser is 100W. Given you are trying to achieve high resolution machining, what was the motivation for using a fibre laser and not a femtosecond laser system? Likewise, rather than using a scanning beam, there are other options for spatially shaped beams. There is a vast literature on SLMs and DMDs for beam shaping, particularly with femtosecond laser machining. It might therefore seem quite important to compare the resolutions that you achieve, with the existing literature in this area?

[Our response 3] We greatly appreciate the reviewer’s very helpful comments about the laser patterning process. As the reviewer knows, the price of fiber laser system is much cheaper than other laser equipment systems, such as femtosecond laser, UV laser, and He-Ne laser systems. That’s why we employed the fiber laser system to show the additional patternability of the first patterned nano-to-micro-structures. We totally agree with the reviewer that the use of selective laser melting (SLM) and electron beam melting (EBM) systems with femtosecond laser machining are very helpful for effective formation of well-defined complex patterns. In response to the reviewer’s comments, we added the suggested references and following sentences to compare the patterning results with existing literatures.

[Modification of the manuscript]

“The sides of square-shaped µm-scale units (81 cells) and mm-scale units (9 cells) are 300 µm and 2 mm, respectively. The laser beam-based processing presented in this study enabled reliable micro-scale patterning. It should be noted that the um-scale pattern structures for the realization of plasmonic metasurfaces or optical device applications can be also manufactured by other programmable beam shaping, such as electron beam lithography [52, 53] and femtosecond laser lithographic process [54, 55]. Although the ytterbium fiber laser-based micromachining system used in this study can achieve a resolution of ~40 µm, we designed unit cells with 300 µm size because the purpose of the fabrication in this study was to make a micro-patterned area that can be visible to the naked eye.” on line 193 of the revised manuscript.

“52. Liang, H.; Lin, Q.; Xie, X.; Sun, Q.; Wang, Y.; Zhou, L.; Liu, L.; Yu, X.; Zhou, J.; Krauss, T. F; Li, J. Nano Lett. 2018, 18, 4460-4466.

  1. Nottola, A.; Gerardino, A.; Gentili, M.; Fabrizio, E. D.; Cabrini, S.; Melpignano, P.; Rotaris, G. Fabrication of semi-continuous profile diffractive optical elements for beam shaping by electron beam lithography. Microelectron. Eng. 2000, 53, 352-328.
  2. Cheng; J. Gu, C.; Zhang, D.; Chen, S.-C. High-speed femtosecond laser beam shaping based on binary holography using a digital micromirror device. Opt. Lett. 2015, 40, 4875-4878.
  3. Wang, X.; Yu, H.; Li, P.; Zhang, Y.; Wen, Y.; Qiu, Y.; Liu, Z.; Li, Y.; Liu, L. Femtosecond laser-based processing methods and their applications in optical device manufacturing: A review. Opt. Laser Technol. 2021, 135, 106687.” on line 390 of the revised manuscript.

[Comment 4] Can you talk about some of the manufacturing applications here, and state how your fabrication method unlocks new capability? Specifically, what samples/features does your fabrication approach allow, that currently is not achievable by other methods? I don’t think I saw a comparison of resolutions and size scales, and fabrication speed, etc. that compares your work with existing work in the literature.

[Our response 4] We appreciate the reviewer’s helpful comments. The combined patterning method of nTP and laser micromachining suggested in this study can realize the multiscale (nano-to-micro-to-millimeter scale) pattern formation on the various surfaces, such as spin-coated soft PMMA thin film, rigid PET film, SUS plate, and Si substrate. We expect that this multiscale patterning strategy can be applicable to various micro- and nano-device fabrications that require complex nano-to-micro pattern geometries within the device systems. For example, the patterned nanoscale, memristive cross-bar array can be patterned to obtain the individual memory cells. (Please see the Figure about the cross-bar resistive memory device below.) In the near future, we will try to develop more effective patterning processes based on the advanced laser systems the reviewer suggested. In response to the reviewer’s comments, we added the following sentences to provide the details of these results.

Figure. High-density memristive crossbar NiOx/Pt nanoarrays via the T-nTP process on a flexible and transparent PET substrate. (C) Schematic illustration of the resistive memory device structure and its resistive switching mechanism through the formation of a Ni filament within NiOx nanowire. (D) High-density NiOx/Pt crossbar resistive memory device created by the T-nTP process. (Top left and top middle) Top-view SEM images, (top right) top-view transmission electron microscopy (TEM)-energy-dispersive spectrometry (EDS) elemental mapping image, (bottom left) cross-sectional TEM-EDS image, and (bottom right) an electron energy-loss spectroscopy measurement result for the NiOx line structure. (E) I-V curve of the NiOx/Pt memristive structure. Scale bars: 100 nm (A), 1 μm [(B) and top left in (D)], and 50 nm [top center in (D)]. (Park et. al., Science advances 6 (31), eabb6462, 2020)

[Modification of the manuscript]

“It should be noted that the um-scale pattern structures for the realization of plasmonic metasurfaces or optical device applications can be also manufactured by other programmable beam shaping, such as electron beam lithography [52, 53] and femtosecond laser lithographic process [54, 55]. Although the ytterbium fiber laser-based micromachining system used in this study can achieve a resolution of ~40 µm, we designed unit cells with 300 µm size because the purpose of the fabrication in this study was to make a micro-patterned area that can be visible to the naked eye.” on line 193 of the revised manuscript.

“In addition, this combined patterning strategy for the realization of high-resolution multiscale pattern geometries that cannot be achieved with a single process will be applicable to various device systems that require complex nano-to-micro structures within device systems.” on line 228 of the revised manuscript.

“52. Liang, H.; Lin, Q.; Xie, X.; Sun, Q.; Wang, Y.; Zhou, L.; Liu, L.; Yu, X.; Zhou, J.; Krauss, T. F; Li, J. Nano Lett. 2018, 18, 4460-4466.

  1. Nottola, A.; Gerardino, A.; Gentili, M.; Fabrizio, E. D.; Cabrini, S.; Melpignano, P.; Rotaris, G. Fabrication of semi-continuous profile diffractive optical elements for beam shaping by electron beam lithography. Microelectron. Eng. 2000, 53, 352-328.
  2. Cheng; J. Gu, C.; Zhang, D.; Chen, S.-C. High-speed femtosecond laser beam shaping based on binary holography using a digital micromirror device. Opt. Lett. 2015, 40, 4875-4878.
  3. Wang, X.; Yu, H.; Li, P.; Zhang, Y.; Wen, Y.; Qiu, Y.; Liu, Z.; Li, Y.; Liu, L. Femtosecond laser-based processing methods and their applications in optical device manufacturing: A review. Opt. Laser Technol. 2021, 135, 106687.” on line 390 of the revised manuscript.
